# Structural insights into the mechanism of archaellar rotational switching

Florian Altegoer [1,7✉], Tessa E. F. Quax [2,3], Paul Weiland [1], Phillip Nußbaum[2], Pietro I. Giammarinaro [1], Megha Patro[2], Zhengqun Li[2], Dieter Oesterhelt[4], Martin Grininger [5], Sonja-Verena Albers [2] & Gert Bange [1,6✉]

Signal transduction via phosphorylated CheY towards the flagellum and the archaellum involves a conserved mechanism of CheY phosphorylation and subsequent conformational changes within CheY. This mechanism is conserved among bacteria and archaea, despite substantial differences in the composition and architecture of archaellum and flagellum, respectively. Phosphorylated CheY has higher affinity towards the bacterial C-ring and its binding leads to conformational changes in the flagellar motor and subsequent rotational switching of the flagellum. In archaea, the adaptor protein CheF resides at the cytoplasmic face of the archaeal C-ring formed by the proteins ArlCDE and interacts with phosphorylated CheY. While the mechanism of CheY binding to the C-ring is well-studied in bacteria, the role of CheF in archaea remains enigmatic and mechanistic insights are absent. Here, we have determined the atomic structures of CheF alone and in complex with activated CheY by X-ray crystallography. CheF forms an elongated dimer with a twisted architecture. We show that CheY binds to the C-terminal tail domain of CheF leading to slight conformational changes within CheF. Our structural, biochemical and genetic analyses reveal the mechanistic basis for CheY binding to CheF and allow us to propose a model for rotational switching of the archaellum.

[1] Philipps-University Marburg, Center for Synthetic Microbiology (SYNMIKRO) & Faculty of Chemistry, Karl-von-Frisch-Straße 14, 35043 Marburg, Germany. [2] Archaeal virus-host interactions, Faculty for Biology, University of Freiburg, Schaenzlestrasse 1, 79104 Freiburg, Germany. [3] Biology of Archaea and Viruses, Groningen Biomolecular Sciences and Biotechnology Institute, University of Groningen, Nijenborgh 7, 9747 AG Groningen, The Netherlands. [4] Max-Planck Institute for Biochemistry, Am Klopferspitz 18, 82152 Martinsried, Germany. [5] Goethe University Frankfurt, Institute of Organic Chemistry and Chemical Biology and Buchmann Institute for Molecular Life Sciences, Max-von-Laue-Straße 15, 60438 Frankfurt am Main, Germany. [6] Max-Planck Institute for terrestrial Microbiology, Karl-von-Frisch-Straße 10, 35043 Marburg, Germany. [7] Present address: Heinrich-Heine University Düsseldorf, Institute of Microbiology, Universitätsstraße 1, 40225 Düsseldorf, Germany. ✉email: altegoer@hhu.de; gert.bange@synmikro.uni-marburg.de

The ability to move towards favorable conditions and away from unfavorable ones is a key feature of microorganisms and enables them to rapidly respond to changes in the environment. For this purpose, bacteria and archaea use two distinct motility structures termed flagella and archaella, respectively, that both generate propulsion through rotational forces[1–4]. However, despite serving a similar purpose, the composition and assembly mechanism are fundamentally different between these two nanomachines[2]. Numerous studies in the recent past have shown that the architecture of the archaellum is rather related to that of type IV pili[5]. Type IV pili are cell appendages that can be found in both archaea and bacteria and in case of the latter are a prerequisite for twitching motility[6,7]. In contrast, archaella rotate to generate propulsive forces, similar to flagella, and thus can be considered rotating type IV pili[5]. The rotation is energized by either the proton motive or sodium motive force for the flagellum or ATP-hydrolysis in the case of the archaellum[8–10].

Given these differences in ancestry and composition, it appears plausible that the regulation underlying directed movement also shows some substantial differences among members of these two domains of life. Curiously however, the chemotaxis machinery is conserved among both bacteria and archaea[11,12]. The chemotaxis system is ubiquitously present in motile archaea belonging to the Eury- or Thaumarchaeota[12–14]. It is responsible for sensing and transferring environmental signals to the so-called 'switch complex' at the motor of the bacterial and archaeal motility structure[11]. In both, euryarchaeal and bacteria, the chemotaxis system and the motility structure are both essential for directional movement. In both systems, the chemotactic signals are perceived by membrane-embedded methyl-accepting proteins (MCP's) that dimerize upon signal recognition leading to autophosphorylation of CheA via the adaptor CheW[15–19]. These three proteins together are organized in large chemosensory arrays that form paracrystalline sheets, which are important for signal integration and amplification[20–22]. The phosphate is then transferred from CheA to CheY, leading to the "active" form of CheY, CheY-P[23,24]. Upon phosphorylation of CheY, a tyrosine and threonine residue are displaced resulting in a shift within the β4-α4 loop and adjacent regions at CheY[25–27]. The critical determinants of this process, an aspartic acid and the coordination of a $Mg^{2+}$ ion are conserved among both archaea and bacteria[26].

In bacteria, these structural rearrangements increase the affinity of CheY-P towards the N-termini of FliY and/or FliM, two proteins that reside within the switch complex of the flagellum (C-ring)[28,29]. The specific binding site includes a conserved 'EIDALL' motif present in FliM and FliY's of all motile bacteria[25,30,31]. The C-ring is formed by the three proteins FliG, FliM, and FliN(Y) and assembles into a cup-like structure at the cytoplasmic face of the flagellar basal body[32,33]. FliG connects the central MS-ring formed by FliF to the motor and stator components[34,35], while FliM and FliY are involved in accepting CheY-P thus leading to conformational changes within the C-ring followed by a directional change in flagellar rotation[36–38]. CheY-P is rapidly dephosphorylated through its autophosphatase activity[39] but also through the phosphatase activity of FliM and FliY in some bacterial species[38,40]. This rapid CheY recycling ensures a fast and reliable response towards chemotactic signals.

In archaea, CheY-P binds to CheF, an adaptor protein uniquely present in archaeal species, however the precise mechanism of this interaction is unclear to date[26,41,42]. CheF resides at the cytoplasmic face of the archaellum switch complex, formed in addition by the proteins ArlCDE[43]. Binding of CheY-P to CheF induces a rotational switch of the archaellum[26,41]. While the mechanism of CheY binding to the switch complex is well-studied in bacterial systems, neither the interaction between CheY-P and CheF nor the conformational changes underlying the

directional change in archaellar rotation have been characterized so far. In an earlier study we could demonstrate that the CheY phosphorylation and subsequent conformational changes within CheY-P are conserved between archaea and bacteria[26]. However, we were unable to retrieve a stable complex between CheY and the adaptor protein CheF.

Here we present the crystal structure of CheF alone and in complex with activated CheY. CheF forms a twisted homodimer offering two CheY-P binding sites within an elongated C-terminal tail domain. Notably, both molecules of the CheF dimer contribute to the interaction interface towards each of the CheY molecules. Two N-terminal pleckstrin homology (PH)-like domains at CheF likely serve as interaction platforms towards the archaeal C-ring. We furthermore deliver a model for rotational switching involving the twisted architecture of the CheF homodimer together with slight conformational changes upon CheY-P binding. Our in vitro data obtained with proteins from the anaerobic euryarchaeon *Methanococcus maripaludis* are supported by in vivo with data from *Haloferax volcanii*, which possesses an advanced euryarchaeal genetic systems and is suited for light microscopy[44,45]. Our study suggests that CheY does not simply represent a "plug-and-play" device to connect chemosensory arrays to the archaellum but instead is an elegant example of co-evolution between two proteins allowing adaptation of an existing system to a different purpose through subtle changes. Thus, our integrative approach delivers the mechanistic basis of how chemotactic signals are transmitted towards the archaellum resulting in rotational switching.

## Results

**CheF forms a twisted homodimer.** To elucidate the mechanistic details of the CheY-CheF interaction, we first sought to gain insights into the molecular architecture of CheF alone. Thus, we employed several archaeal homologs of CheF to solve its structure by X-ray crystallography. Notably, structure solution of CheF has been attempted earlier using a homolog from the hyperthermophilic archaeon *Pyrococcus horikoshii*[46]. We solved the crystal structure of CheF from *Methanocaldococcus jannaschii* (Table S1) by selenium single-wavelength anomalous diffraction (Se-SAD), which allowed us to build an initial model of the N-terminal domain of CheF. Unfortunately, a translational non-crystallographic symmetry combined with a moderate resolution of 3.6 Å complicated model building. However, we could successfully use the dataset deposited by Paithankar and coworkers[46], to solve the structure of the *P. horikoshii* at 2.75 Å by molecular replacement (MR) using our initial *Mj*CheF model and obtained a complete model of CheF. Through this approach, we were also able to include native phases obtained by Se-SAD to assist model building during refinement (Table S1).

CheF consists of two N-terminal domains and a long C-terminal tail that reaches out from the center of the two domains and extends into a long tail domain of over ~80 Å in length (Fig. 1a, b). The first N-terminal domain is formed by seven β-strands (i.e., β1-β7) and an α-helix and connecting it to the second domain (Fig. 1b). This second N-terminal domain consists of eight β-strands (i.e., β8-β15) and is connected to a small helical bundle (i.e., α2-α4) central to the two N-terminal domains. Helix α4 spans into an elongated C-terminal domain (CTD) that is formed by α5-α7, a long β-strand (i.e., β16) and the C-terminal helix α8. The asymmetric unit contained two CheF molecules, which interacted through their C-termini employing a buried surface area of 2700 $A^2$ (Fig. 1c). The strongest determinant of this interface is a substantial β-sheet, jointly being formed by the β16 strands of each of the CheF monomers (Fig. 1c and S1a). Moreover, both monomers of the CheF

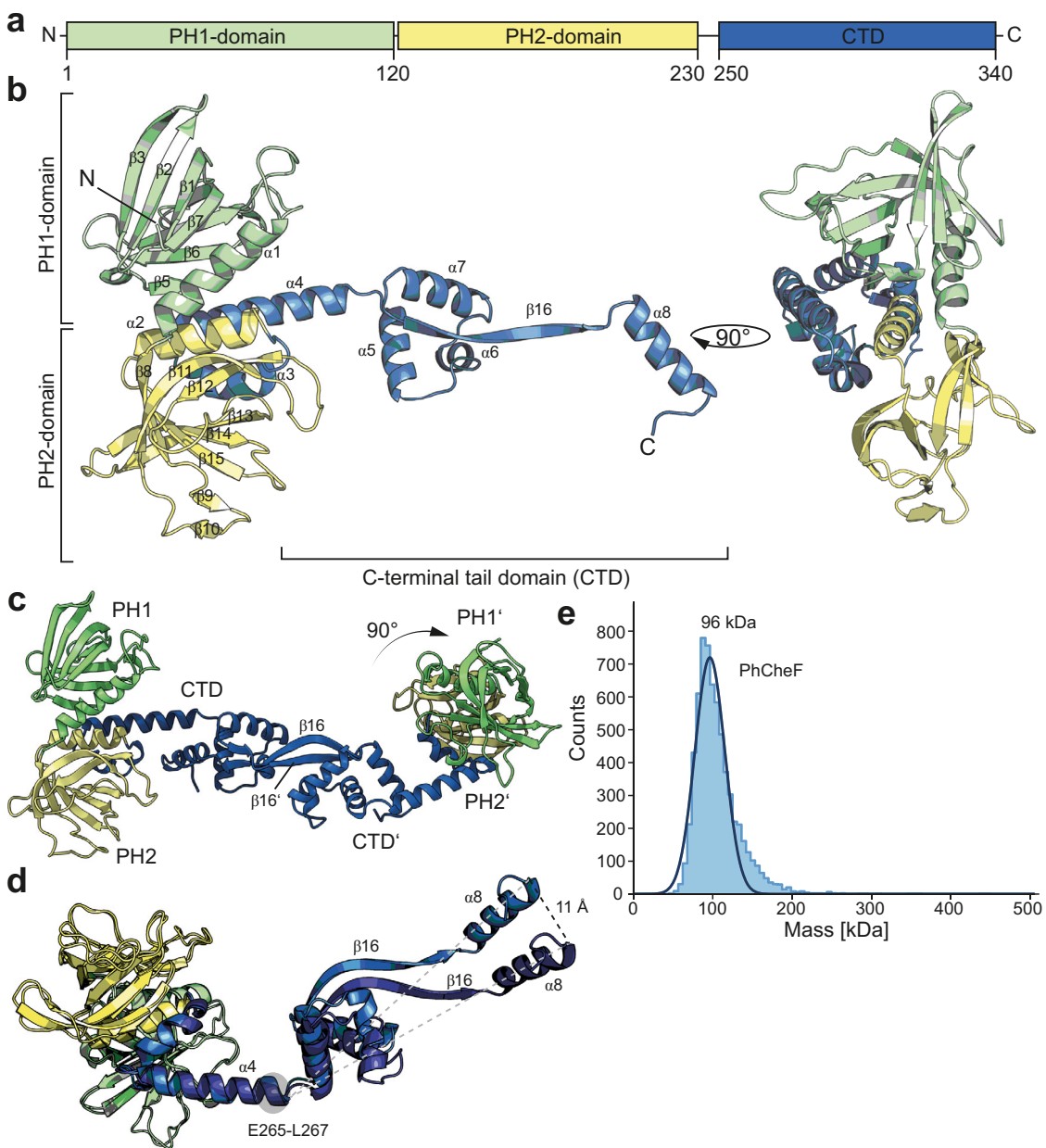

**Fig. 1 The crystal structure of CheF reveals a twisted homodimer. a** Domain architecture including amino acid boundaries of CheF. **b** Crystal structure of PhCheF with the two PH-domains colored in light green and light yellow and the CTD in blue, respectively. The right panel shows the structure rotated by 90°. **c** The asymmetric unit of the PhCheF structure reveals the presence of a dimer. The two monomers are twisted via their CTD's resulting in the PH domains being rotated by 90°. **d** The two monomers of the CheF dimer exhibit inherent flexibility through a region within helix α4. **e** Mass photometry of PhCheF shows a single species of ~96 kDa. Source data are provided as a Source Data file.

homodimer do not align symmetrically to each other but instead tilt their N-terminal domains by 90° (Fig. 1c). Superposition of the two monomers using N-terminal residues 2-260 (r.m.s.d = 0.8 Å) shows this rotational movement of the CTD (Fig. 1d). The residues involved in bending are E265 to L267 within helix α8 (obtained from DynDom[47]). This observation suggests that the CheF homodimer exhibits an intrinsic flexibility with respect to its N-terminal domains, which might be of functional relevance. Thus, our crystal structure suggests a CheF homodimer, which was confirmed in solution by mass photometry (MP). We observed a single species of 96 kDa that included 89 % of all measured observations (Fig. 1e). Furthermore, we investigated the oligomerization behavior of two CheF homologs from *M. maripaludis* (MmCheF) and *Thermococcus kodakaraensis*

(TkCheF). The two proteins also formed dimers in solution as judged from the molecular weight determined by MP of 88 kDa and 82 kDa for MmCheF and TkCheF, respectively (Fig. S1a). Judging from the overall high sequence identity of CheF homologs, our data thus suggest that dimer formation is conserved among CheF's (Fig. S1b). Taken together, our structural analysis reveals two molecules of CheF forming an elongated twisted homodimer via their tightly connected C-terminal tail domains.

A DALI-search[48] revealed that the N-terminal domains show a high structural similarity to pleckstrin homology (PH) domains known from e.g. signaling proteins[49]. The PH domain was initially described in the protein kinase C (PKC) substrate pleckstrin and several proteins involved in signal transduction

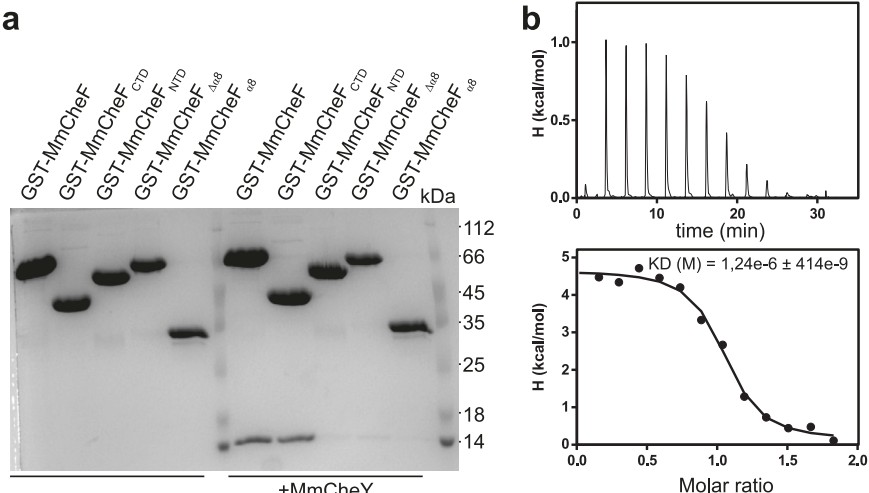

**Fig. 2 CheY-P interacts with the CTD of CheF. a** Coomassie-stained SDS-PAGE of a GST-interaction assay employing GST-tagged versions of MmCheF and CheY. Representative image derived from three independent experiments. **b** Isothermal titration calorimetry (ITC) of MmCheY and GST-MmCheF$_{CTD}$ in presence of BeF$_3^-$ yielding a $K_d$ of 1.24 ± 0.414 µM. GST-MmCheF$_{CTD}$ was added to the sample cell and titrated with MmCheY. The black dots represent the ΔH per injection of titrant into the cell and the solid line represents the fitting curve for all recorded injections. Source data are provided as a Source Data file.

processes[50,51]. Both N-terminal domains superpose reasonably well to selected PH-domains with a root mean square deviation (r.m.s.d) of 2.8 Å over 70 Cα-atoms (Fig. S2a). A closer inspection of the potential phosphate-binding site at PhCheF indicates that phosphate binding is unlikely within the two CheF domains as judged by overall polarity and the lack of coordinating residues (Fig. S2b). Most likely, these PH domains serve as platforms for protein-protein interactions within CheF as reported by several recent studies on PH domains[49].

**Activated CheY interacts with the C-terminal tail domain of CheF.** With the structural information at hand, we aimed to understand how phosphorylated CheY (CheY-P) might bind to CheF. To identify the potential binding site of CheY-P at CheF, several truncated constructs of *M. maripaludis* CheF fused to an N-terminal GST-tag were generated as summarized in Fig. S3a. GST-Interaction assays employing these CheF constructs were performed in the presence of 2 mM BeF$_2$ and 20 mM NaF to mimic a phosphorylated state of CheY (compare also to Quax et al.[26]). The full-length GST-CheF showed a specific interaction with CheY-P, which was not impacted by deletion of the two N-terminal PH-domains (MmCheF$_{CTD}$; Fig. 2a). Truncation of the CTD (MmCheF$_{NTD}$) abolished CheY-P binding and deleting the C-terminal α-helix (α8) also abolished CheY-P binding (Fig. 2a). Notably, a GST-MmCheF$_{α8}$ construct was also not sufficient to recruit CheY-P (Fig. 2a). Thus, we conclude that presence of the complete CTD of CheF is required for its interaction with CheY-P.

To substantiate our findings, we employed isothermal titration calorimetry (ITC) to determine the dissociation constant between CheF$_{CTD}$ and CheY-P, which was mimicked through the presence of BeF$_2$ (see previous section). Here, we employed the GST-MmCheF$_{CTD}$ construct used for the interaction assays at it showed a higher stability compared to a construct lacking the GST-tag. As GST forms dimers, we investigated a potential oligomerization of the fusion protein by MP. Our data show that the main fraction (74%) has a molecular weight of 76 kDa indicating a dimer, while only 20 % of the molecules had a molecular weight of 152 kDa indicating a tetramer (Fig. S3b). We obtained a $K_D$ of 1,3 ± 0,4 µM (Fig. 2b). Our titration curve shows that the binding reaction is endothermic. Notably, a weak binding

of CheY to CheF could also be observed in the absence of BeF$_3^-$ as indicated by slight thermal shifts upon titration of CheY (Fig. S3c). Our titration curve suggested that CheY-P interacts in a 1:1 stoichiometry with CheF. We therefore assume that 2 molecules of CheY-P will most likely interact with one CheF dimer (see also below).

**The CTD of CheF is important for directional movement of *H. volcanii*.** To study the physiological role of the CheF CTD, we expressed the CheF protein in the halophilic euryarchaeon *H. volcanii*. We fused GFP both to the C- or the N-terminus of CheF and cloned these under a tryptophan inducible promoter on a plasmid. GFP-tagged CheF mainly localized as distinct foci at the poles of the rod-shaped *Haloferax* cells (in ~55% of cells) (Fig. 3a). Foci were found either at one or at both cell poles, corresponding with previous findings on the cellular positioning of CheF[43,45]. It has been shown that CheF localization is normally depended on the archaellum motor, where it requires a complex of the archaellum proteins ArlCDE to dock to the motor[43]. Neither did the C- nor did the N-terminal GFP fusions affect the percentage of cells with polar foci (Fig. 3a). These findings indicate that the GFP fusion does not influence the binding of CheF to the cytoplasmic side of the archaellum (Fig. 3a).

Next, we used semi-solid agar plates to assess whether the GFP fusions of CheF would be affected in their biological role, which is the switching of the motor rotation of the archaellum. Wild type *H. volcanii* cells form motility rings on semi-solid agar plates in several days. A Δ*cheF* strain with an empty plasmid (pTA1228) cannot form these motility rings anymore. Expression of N-terminally GFP-tagged CheF (GFP-CheF) in the Δ*cheF* strain restored the phenotype and motility rings of similar diameter as the wild type were formed (Fig. 3b, c). In contrast, the C-terminally GFP tagged CheF (CheF-GFP) did not complement the Δ*cheF* phenotype and no motility rings were observed. This indicates that when a GFP is present at the C-terminus of CheF, the biological function of the protein is compromised. It was previously determined that CheF can interact both with the archaellum motor proteins ArlCDE and with the response regulator CheY[26,41,43]. The analysis with fluorescent microscopy showed that interaction with ArlCDE is still intact, suggesting

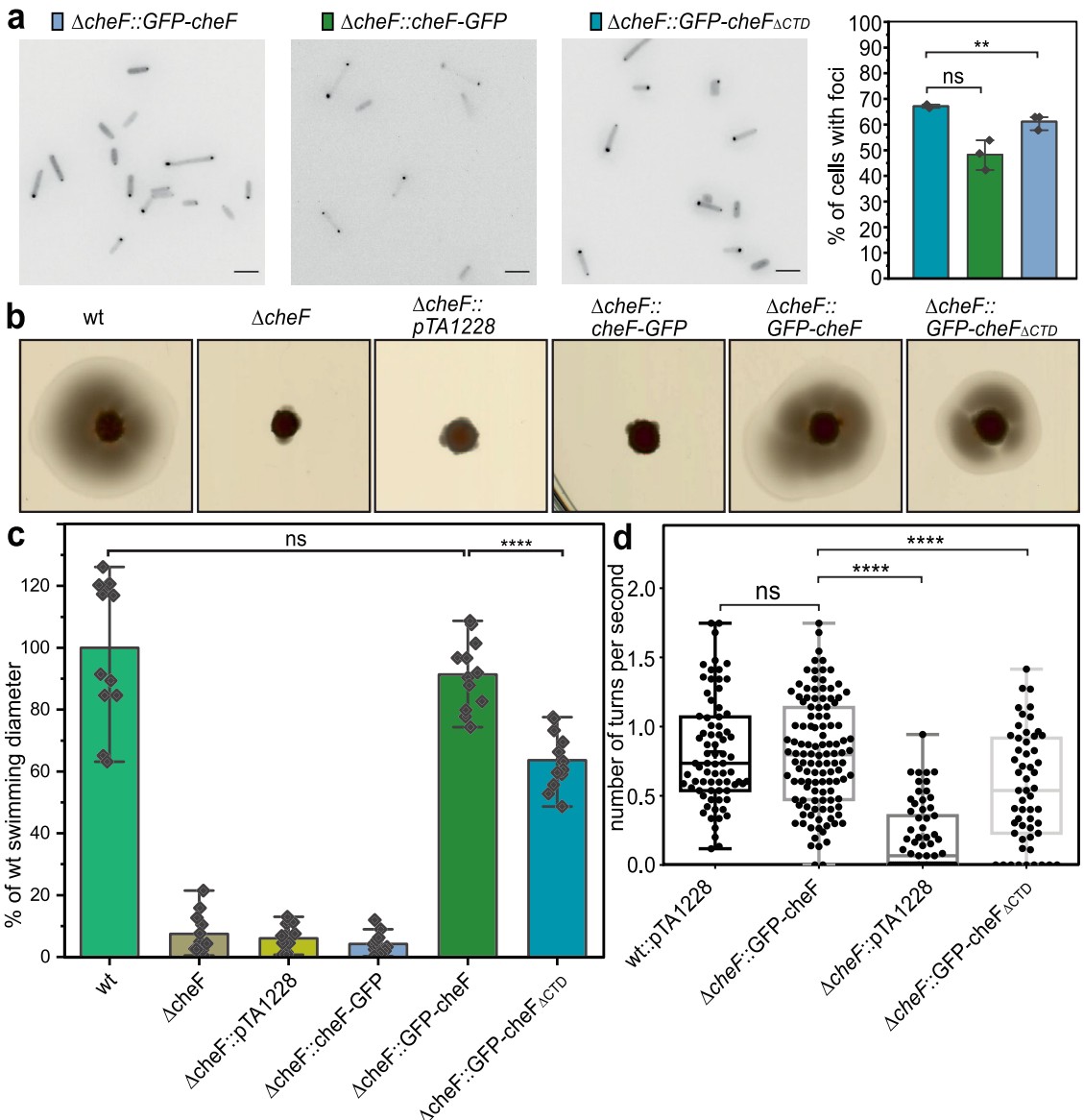

**Fig. 3 Motility behavior of different CheF mutants in _H. volcanii_. a** Representative fluorescent images of intracellular distribution of GFP labeled CheF protein in the Δ_cheF_ strain are shown. Scale bars, 4 μm. (right panel) The percentages of cells with intracellular CheF foci in the 3 strains are shown. Data are represented as mean values ± standard deviation. The dots indicate the data from three biological replicates. Δ_cheF_::_cheF-GFP_, n = 2569 cells, Δ_cheF_::_GFP-cheF_, n = 1570 cells, Δ_cheF_::_GFP-cheF_ΔCTD, n = 413 cells. _cheF_ΔCTD, encoding CheF protein with 83 aa C-terminal truncation. ns, not significant as determined by unpaired two-sided T-test (p = 0.052). **P < 0.01 (p = 0.006). **b** Motility rings of different _H. volcanii_ strains on semi-solid agar plates made of YPC medium. **c** Quantification of the diameter of the motility rings such as shown in **a**. The experiment was performed with at least three technical and two biological replicates (technical replicates are necessary because of the different humidity conditions in individual plates). WT, _H. volcanii_ H26; Δ_cheF_, _H. volcanii_ H26 deleted for _cheF_; pTA1228, empty plasmid; _cheF_ΔCTD, encoding CheF protein with 83 aa C-terminal truncation. ns, not significant (p = 0.251). ****P < 0.0001 as calculated with unpaired two-sided T-test. Data are represented as mean values ± standard deviation. n = 13 experiments **d**. Swimming behavior of different _H. volcanii_ strains expression a truncated version of CheF. Time lapse movies of swimming cells in liquid medium were analyzed and the average swimming behavior of individual cells is displayed in a box and whiskers plot. The whiskers show the minimum and maximum. The hinges of the box represent the 25–75 percentile and the middle line in the box is the median of all values. The frequency of reversals is measured as the time between two subsequent reversals (angle >90°). ****P < 0.0001 significantly different as established with unpaired two-sided T-test. ns, not significant as established by unpaired two-sided T-test, (p = 0.371). The wt strain in the analysis is Δ_pyrE2_ (H26). pTA1228 empty plasmid. wt::pTA1228, n = 78 cells, Δ_cheF_::_GFP-cheF_, n = 120 cells, Δ_cheF_::pTA1228, n = 81 cells, Δ_cheF_::_GFP-cheF_ΔCTD, n = 55 cells. Source data are provided within the Source Data file.

that the C-terminal GFP fusion specifically blocks interaction with CheY-P (Fig. 3a).

To further analyze the function of the C-terminus of CheF, we constructed two C-terminal truncations of CheF, in which either the complete C-terminus (83 aa, GFP-CheFΔCTD) or only the 8th alpha helix (14 aa, GFP-CheFΔα8) was deleted. Both constructs were cloned with an N-terminal GFP fusion under a tryptophan

inducible promoter and transformed to the Δ_cheF_ strain. Analysis with fluorescent microscopy showed that the positioning pattern of GFP-CheFΔCTD was reminiscent of that of full-length GFP-CheF, indicating that the C-terminus is not involved in binding to the archaellum motor (Fig. 3a). Expression of GFP-CheFΔα8 resulted in diffuse fluorescence in the cytoplasm in all analyzed cells (Fig. S3c, d). As the C-terminus is not required for correct

cellular positioning of CheF, we assume that the deletion of α8 in *H. volcanii* renders the protein unstable and that this truncated protein it is not correctly expressed or folded. This is supported by our western blot analysis of the CheF-GFP constructs showing less full-length protein of GFP-CheF$_{\Delta\alpha8}$ compared to the other CheF-GFP constructs (Fig. S4). In correspondence to that, we observed that GFP-CheF$_{\Delta\alpha8}$ expression does not yield motility rings at all (Fig. S3d).

Motility assays on semi-solid agar plates showed that the Δ*cheF* strain expressing GFP-CheF$_{\Delta CTD}$ did form motility rings. However, their diameter was significantly reduced in comparison with expression of full-length GFP-CheF (~30%). A C-terminal GFP fusion, thus leads to a more severe phenotype (complete absence of motility rings) in comparison with C-terminal truncated CheF (motility rings of reduced diameter) (Fig. 3b, c). The C-terminal GFP fusion and the C-terminal truncation of CheF, both show that the C-terminus is indeed involved in interaction with CheY-P. To study this interaction in more detail, time lapse microscopy was applied to study the swimming behavior of the various mutant strains. Wild type *H. volcanii* cells display runs of forward and reverse swimming that are randomly alternated in the absence of stimuli[52,53]. Indeed, when cells of the background strain H26 with an empty plasmid were analyzed (wt::pTA1228), we observed frequent reversals (defined as >90° turns) and on average cells made 0.82 reversals per second (Fig. 3d). A Δ*cheF* strain expressing GFP-CheF from a plasmid had on average the same reversals per second as the wild-type strain, indicating that this plasmid is capable of correct complementation of the cheF deletion. When an empty plasmid was expressed in the Δ*cheF* strain, only 0.2 reversals per second were observed and the cells were mainly swimming smoothly without many reversals (Fig. 3d). This phenotype is similar as to what was observed previously for a Δ*cheF* strain in *H. volcanii*[26]. Next, the swimming behavior of the GFP-CheF$_{\Delta CTD}$ was compared with that of full-length GFP-CheF when expressed in the Δ*cheF* strain. In this case, a significant reduction of the number of reversals was observed compared with the full-length CheF (~0.55 vs 0.82 reversals per second) (Fig. 3d). The effect was not as strong as in the complete absence of CheF (~0.2 reversals per second), which corresponds to the efficiency of directional movement as observed on semi-solid agar plate.

In summary, deletion of the CheF CTD is resulting in a strain with a reduced directional movement and altered swimming behavior in *H. volcanii*. Thus, the C-terminus of *H. volcanii* CheF is important for directional movement, due to its role in determination of the archaellum rotational direction.

**Both CheF molecules of the homodimer contribute to CheY-P binding**. All attempts to gain a more detailed molecular picture of the CheY-CheF interface were unsuccessful as we could not reconstitute a size-exclusion stable CheY-CheF complex employing the full-length proteins. We therefore constructed a fusion protein consisting of *M. maripaludis* CheY and the C-terminal 103 residues of CheF separated by a 20 amino acid linker including a thrombin cleavage site (Fig. 4a; see material and methods for details). The fusion protein was produced in *E. coli* and migrated in a stable fraction on SEC with a molecular weight of $60 \pm 10$ kDa confirmed by multi-angle light scattering (MALS) (Fig. S5a, left panel). We repeated the SEC-MALS analysis in the presence of 2 mM BeF$_2$ and 20 mM NaF. Interestingly, the mass determination was more accurate and resulted in a determined molecular weight of $50 \pm 5$ kDa (Fig. S5a, right panel). Most likely, activation of CheY stabilizes the interaction with the CheF$_{CTD}$ and leads to slight conformational changes resulting in a more compact particle.

We crystallized the CheY:CheF$_{CTD}$ in the presence of 2 mM BeF$_2$ and 20 mM NaF (for more details, see Materials and methods section) and could solve the structure of this activated CheY:CheF$_{CTD}$ to a resolution of 2.3 Å (Table S1), which allowed us to model both the full CheY-P and 70 amino acids of the CheF$_{CTD}$ into the electron density. A CheF$_{CTD}$ dimer bound to two CheY-P molecules was observed in the asymmetric unit (Fig. 4b). The interaction interface between CheY-P and CheF$_{CTD}$ involved the three helical bundle at CheF$_{CTD}$ with one molecule contributing α5 and α6 and the other one α8, covering a total of 778 Å$^2$ buried surface area (Fig. 4c). At CheY-P, the interface mainly involves the regions directly adjacent to the BeF$_2$ binding site, namely α1 and the β3-α4-loop as well as the β4-α5-loop (Fig. 4c). Comparing CheY-P to its non-activated form revealed that structural rearrangements upon activation mainly occur in the two mentioned loop regions and thus directly adjacent to the interaction interface with CheF (Fig. 4c). At CheF, conformational changes are mostly limited to helix α8 being kinked towards CheY-P upon binding. A closer inspection of the CheY-CheF$_{CTD}$ interaction showed that the interaction is mostly mediated by hydrophobic residues and some polar backbone contacts "clamping" the interface on one side (Fig. S5b). Interestingly, the terminal carboxyl-moiety of MmCheF F348 reaches into the phosphorylation site contacting one of the waters coordinating the Mg$^{2+}$ and MmCheY K107 (Fig. S5b, 5 and S6). In the case of a phosphorylated D57, this carboxyl-moiety might also be involved in dephosphorylation instead of stabilizing the interaction between CheF and CheY-P.

In conclusion, our structural analysis provides insights into the molecular mechanism of CheY-P recognition by the archaeal adaptor protein CheF. We show that two CheY-P molecules bind to the dimeric CTD of CheF involving an interface near the phosphorylation site of CheY thus allowing a direct sensing of the phosphorylation state of CheY by CheF.

## Discussion

**CheY signal transduction: a different binding mode in flagella and archaella?** The exact binding site(s) of CheY-P at FliM and FliY(N) in the flagellum and consequent conformational changes in the C-ring have long been only poorly understood and the molecular mechanisms remained enigmatic. Structural information on a CheY-P/FliM interaction was limited to several CheY structures bound to the N-terminal FliM peptide[25,30]. Our structure of CheY-P bound to CheF enabled us to compare its binding at the base of the archaellar motor with the equivalent situation at the bacterial flagellum. At the bacterial flagellum, CheY-P interacts with the flagellar C-ring through a conserved 'EIDALL' motif present in the first helix of the protein FliM (FliM$_N$)[30]. Thus, we superimposed our archaellar CheY-P/CheF structure with that of the flagellar CheY-P/FliM$_N$[30]. Both CheY-P proteins superpose well with an r.m.s.d. of 1.042 over 604 atoms (Fig. 5). To our surprise, the interaction of CheY-P with its archaellar and flagellar clients CheF and FliM, respectively, differed fundamentally: While FliM$_N$ binds into a surface groove formed by the CheY helices α4 and α5, CheF resides near the BeF$_3^-$ and mainly involves α1 and the loops on top of CheY-P (Fig. 5).

Notably, it was also suggested that the N-terminus of FliM represents only part of the binding interface of CheY-P to FliM. More precisely, Dyer and coworkers identified the middle domain of FliM (FliM$_M$) to interact with a region adjacent to the phosphorylation site at CheY-P[29]. In their study, they report a similar binding site of FliM$_M$ at CheY-P as we observed in our structure of CheY:CheF$_{CTD}$ (compare Fig. 4). In some flagellated

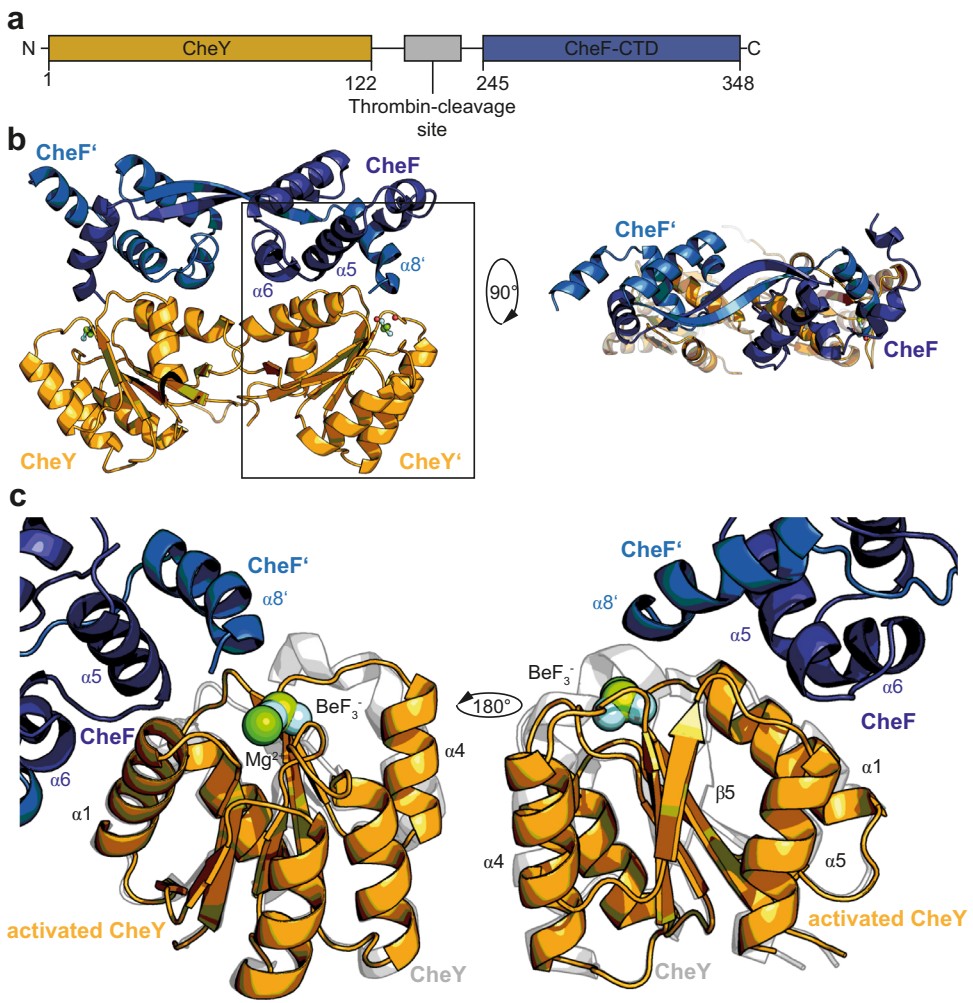

**Fig. 4 CheY-P interaction involves a complex interface at CheF formed by both molecules of the dimer. a** Domain architecture of the MmCheY:CheF$_{CTD}$ fusion construct. **b** Crystal structure of the MmCheY:CheF$_{CTD}$ fusion construct. **c** Closeup of the binding site of CheY at CheF. The interaction interface involves α5, α6 from one CheF molecule and α8 from the other molecule of CheF. MmCheY is colored in orange and the CTD of MmCheF is colored in blue. The non-phosphorylated CheY (PDB: 6EKG) has been superposed and is shown in gray in the background to depict the structural differences between activated- and apo-CheY.

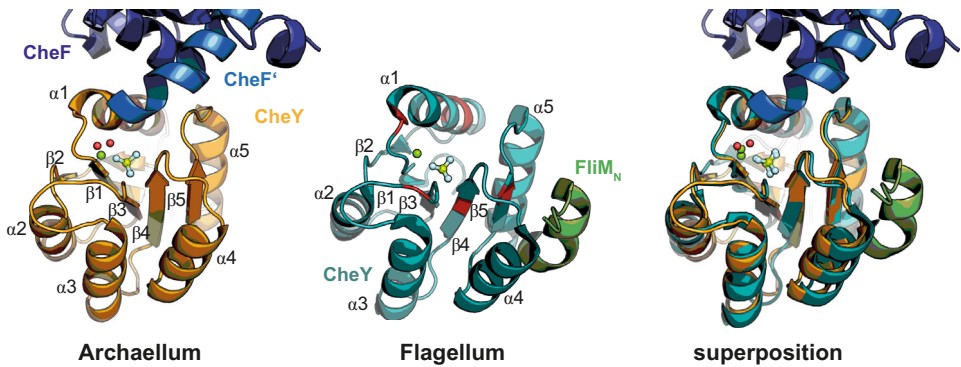

**Fig. 5 Structural comparison between CheY bound to CheF and FliM$_N$.** Superposition of CheY bound to CheF and CheY bound to the N-terminus of FliM (FliM$_N$; PDB: 1F4V). *Ec*CheY is colored in cyan and FliM$_N$ in smudge. Residues identified in ref. [29] important for FliM$_M$ interaction at CheY are colored in red.

bacteria, the interaction between the middle domain of FliM and CheY-P is key to CheY-P recycling, as FliM$_M$ has phosphatase activity and rapidly dephosphorylates CheY-P to allow a rapid response to CheY-P molecules[40]. CheF is lacking a bona fide phosphatase domain, but our structure even suggests that the terminal carboxy group of MmCheF F348 and might be involved in dephosphorylation as it directly reaches into the phosphorylation site and contacts the BeF$_3^-$ phosphomimic via a coordinated

water molecule (Fig. S6). The major interaction site at CheY-P seems to be similar between the flagellar and the archaeal systems despite the strong structural differences of FliM and CheF.

The role of the $FliM_N$ was recently clarified by demonstrating that binding of CheY-P to FliM and FliN(Y) in the flagellar system is actually a "two-step" binding process[54]. The N-terminus of FliM (and FliN(Y)) only serves as a high affinity binding site to increase the number of CheY-P molecules that subsequently bind to the FliM middle domain core[54]. In contrast, we now demonstrate that activated archaeal CheY-P has only one binding site at CheF, with two molecules of CheY-P binding one CheF dimer (compare Fig. 4). In the flagellar system different affinities of CheY-P towards FliM were reported. While Park and coworkers reported a $K_d$ of $39 \pm 5$ nM of activated CheY towards $FliM_{NM}$ (residues 1-249) in *T. maritima*[55], McAdams et al. only observed a $K_d$ of 27 μM of activated CheY towards a $FliM_N$ peptide[39]. The affinity of archaeal CheY-P towards CheF of 1.24 μM is between those observed in the flagellar systems. In conclusion, our analysis shows that CheY-P employs different binding modes at the archaellum and the flagellum although a similar interface at CheY-P is recognized in the two motility systems.

**How is CheF connected to the ArlCDE complex?** The archaeal motor including the C-ring composed of ArlCDE has recently been studied by cryo-electron tomography (cryo-ET) and sub-tomogram averaging revealing the overall architecture of this machinery[56]. The ArlCDE complex forms an assembly with a six-fold symmetry on the inner face of the archaellum arranged in a viaduct-like shape[56]. Despite the diversity of ArlCDE proteins among different archaeal species, the general architecture of this archaellar C-ring is likely conserved among archaeal species[43]. Earlier studies have suggested that CheF interacts with the ArlCDE complex[41,42], which was recently confirmed in vivo[43]. As the dimer of the CheF CTD is occupied by two CheY-P molecules, the specific fold of the two N-terminal PH-domains suggest that they serve as interaction platform towards the ArlCDE complex. Many recent studies have expanded the repertoire of PH-domain containing proteins in a wide range of cellular environments where they serve various purposes conferring protein-protein interactions and to a lesser extent also phospholipid-binding[49–51,57]. Furthermore, our data showed that a deletion of the CheF CTD did not impact its localization towards the pole of *H. volcanii* cells (Fig. 3a).

This idea is further supported by the distance between the PH-domains of each CheF monomer being roughly 100 Å and thus a similar distance as the spacing of the "viaduct leg" density assigned to the ArlCDE complex (Fig. 6a). We thus consider it likely that CheF binds to this distal part of the archaeal switch complex via its PH-domains, while the CTD remains accessible to CheY-P binding.

**Possible implications of CheY-P binding on the CheF conformation.** In bacteria, the switch complex undergoes a substantial remodeling upon CheY-P binding to FliM that changes the rotation direction of the flagellum[58]. With the apo and CheY-bound structures of CheF at hand, we aimed to understand whether structural changes within CheF can also be observed in the archaeal system. As our CheF structures in activated and non-activated states were derived from two different organisms, we first investigated the similarity of the CTD's (Fig. 6b). Despite the sequence similarity of 36%, the domains showed almost identical structural elements (Fig. 6b). Upon superposition, we observed some slight conformational changes as helix α8 was pulled towards the phosphorylation site (Fig. 6c). In both structures,

helix α8 and α4 of the opposing monomers are tightly connected via a hydrophobic interface and polar contacts (Fig. 6c). Thus, the movement of helix α8 displaces helix α4. As our structure of CheF bound to an activated CheY is lacking the N-terminal PH domains, we can only speculate that this movement is further transmitted. However, combined with our observation of the inherent flexibility of the CTD's through a patch in helix α4 (compare to Fig. 1d), we consider it likely that slight conformational changes upon CheY-binding in the CTD would lead to larger changes in the PH-domains. In the clockwise (CW) state, apo CheF would reside on the basal side of the ArlCDE ring (Fig. 6c). Upon binding of phosphorylated CheY, the N-terminal PH-domains at CheF would get displaced, inducing a conformational change of the ArlCDE complex and subsequently result in a change of the rotational direction to counterclockwise (CCW) (Fig. 6c).

Taken together, our study delivers the mechanistic basis of how CheY-P binds to the adaptor protein CheF and allows to propose a model for how rotational switching of the archaellum might be mediated by slight conformational changes within CheF.

## Methods

**Accession numbers.** The protein sequences used in this study are available at NCBI under the following accession numbers: *Methanococcus maripaludis* CheY (WP_011170877 [https://www.uniprot.org/uniprot/Q6LYQ5]), *Methanococcus maripaludis* CheF (WP_181493154 [https://www.uniprot.org/uniprot/A0A7J9P5K9]), *Pyrococcus horikoshii* CheF (WP_010884600 [https://www.uniprot.org/uniprot/O58230]), *Thermococcus kodakarensis* CheF (WP_011249592), *Haloferax volcanii* CheF (WP_004043721 [https://www.uniprot.org/uniprot/Q5JF87]).

**Plasmid generation and construct design.** For the plasmid constructions, standard molecular cloning strategies and techniques were applied[59]. All plasmids and primers used in this study are listed in Tables S2 and S3. For the overproduction of MmCheF, several constructs were generated. The regions encoding the full-length *MmcheF* as well as truncation constructs were amplified by PCR and inserted into the *NcoI/XhoI* restriction sites of the vector pGAT2 which adds an N-terminal Glutathione-S-transferase (GST)-tag to the protein. In addition, the full-length *MmcheF*, *TkcheF* and *PhcheF* sequences were amplified by PCR and inserted into the *NcoI/XhoI* restriction sites of the vector pET24d.

To generate pET24d-MmCheY:CheF$_{CTD}$, *MmcheY* and the respective region of *MmcheF* were amplified by PCR including a linker region encoding a thrombin cleavage site. Both fragments were inserted into the *NcoI/XhoI* restriction sites of the vector pET24d. Plasmids based on pSVA3922[45] and pIDJL-40[60], with *pyrE2* for selection with uracil, were constructed to express GFP-tagged proteins in *H. volcanii* strains (table S4) using primers listed in table S3.

**Protein production and purification.** The different CheF homologs, GST-MmCheF constructs, MmCheY and MmCheY:CheF$_{CTD}$ were produced in *E. coli* BL21 (DE3) (Novagen). The protein production was performed in auto-inductive Luria-Miller broth (Roth) containing 1 % (w/v) α-lactose (Roth). The cells were grown for 20 h at 30 °C and 180 rpm. The cultures were harvested by centrifugation (4,000 g, 15 min, 4 °C), resuspended in HEPES buffer (20 mM HEPES, 200 mM NaCl, 20 mM KCl, 40 mM imidazole, pH 8.0), and subsequently disrupted using a microfluidizer (M110-L, Microfluidics). The cell debris was removed by centrifugation (50,000×g, 20 min, 4 °C). The supernatant was loaded onto Ni-NTA FF-HisTrap columns (GE Healthcare) for affinity purification via the hexahistidine tag. The columns were washed with HEPES buffer (10x column volume) and eluted with HEPES buffer containing 250 mM imidazole. The protein was concentrated with Amicon Ultra-30K centrifugal filters and subjected to SEC using a HiLoad 26/600 Superdex 200 column equilibrated in HEPES buffer without imidazole and a pH of 7.5. The peak fractions were analyzed using a standard SDS-PAGE protocol, pooled, and concentrated with Amicon Ultra-30K centrifugal filters.

**Protein production and purification of SeMet MjCheF.** MjCheF was produced in *E. coli* BL21 (DE3) (Novagen). A pre-culture of 400 ml LB medium was grown for 16 h at 37 °C under constant shaking at 180 rpm. The cells were harvest at 4000×g for 15 min and resuspended in 10 ml M9 medium. The resuspended cells were used to inoculate 5 l of M9 medium (52 mM $Na_2HPO_4$; 24 mM $KH_2PO_4$; 9.5 mM NaCl; 20 mM $NH_4Cl$; pH 7.4; in dd$H_2O$) to an OD600 of 0.1. The M9 medium was infused with sterile and freshly made SolX solution (2 g/l of DL-Lysine, DL-Threonine, and DL-Phenylalanine; 1 g/l of DL-Leucine, DL-Isoleucine, DL-Seleno-Methionine, and DL-Valine; 40 mM $MgCl_2$; 5 mM $CaCl_2$; 80 g/l glucose; in dd$H_2O$; sterile filtered). The cells were grown to an OD600 of 0.6 at 37 °C and 180 rpm. Protein production was induced by adding 1 mM IPTG. The cultures

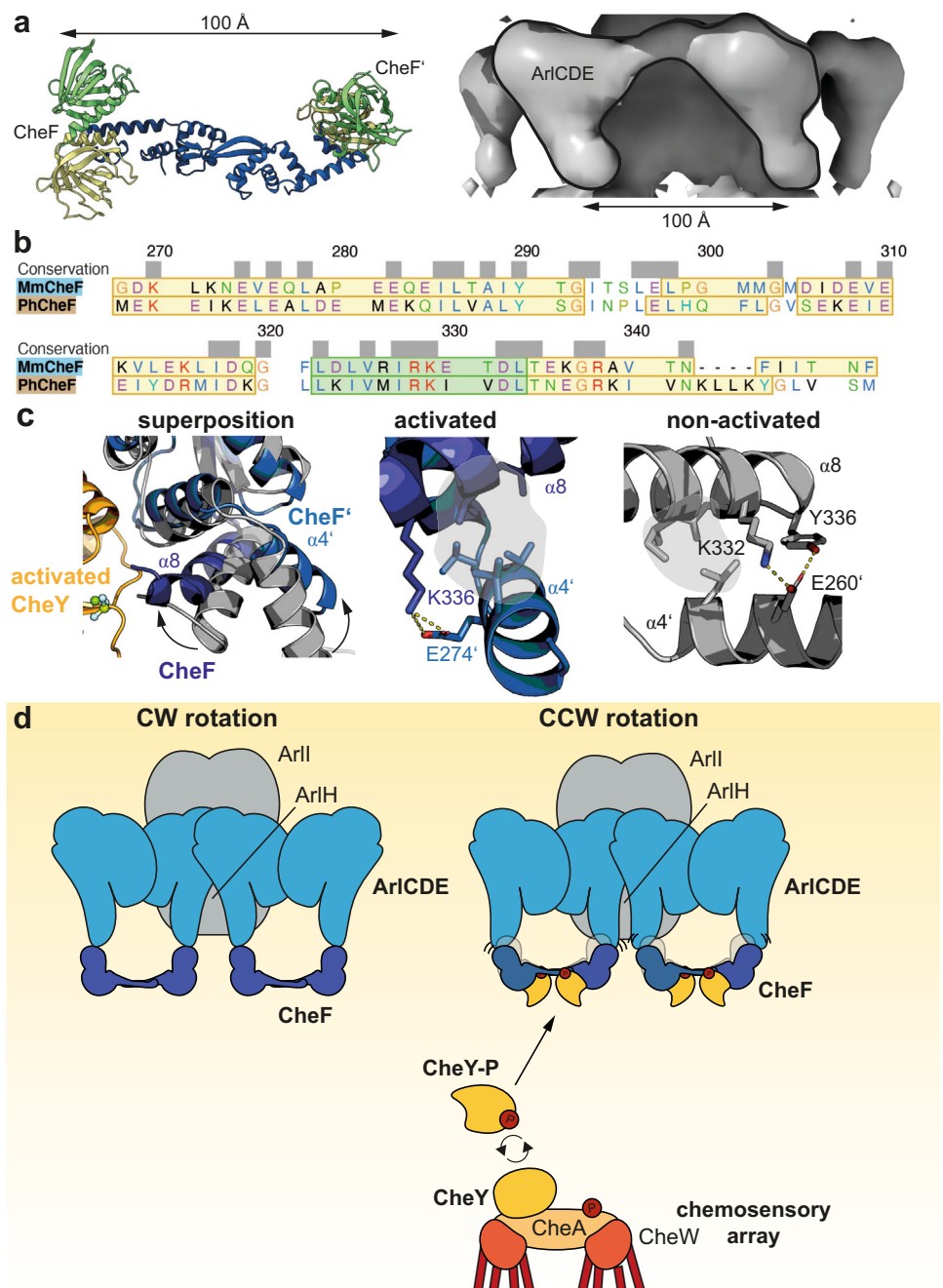

**Fig. 6 Binding of CheY-P to CheF leads to conformational changes. a** Distance between the PH-domains of the two CheF monomers and the "legs" of the ArlCDE-complex assigned density (EMD-3759) is ~100 Å. **b** Sequence alignment of the CTD's of MmCheF and PhCheF. The actual secondary structure elements derived from the two structures are projected onto the amino acid sequence. Yellow boxes mark α-helices and green boxes indicate β-strands. Residues are colored according to the Clustal X coloring scheme that depends on the residue type and conservation pattern in the respective column. **c** Superposition of CheF alone (gray) and bound to activated CheY-P shows conformational changes of helix α8 and subsequent movement of helix α4 through polar and hydrophobic interactions. **d** Model of how rotational switching might occur upon binding of CheY-P to CheF. The twisted architecture combined with the conformational changes at CheF might be transmitted to the ArlCDE complex and lead to slight changes in the C-ring architecture.

continued to grow at 37 °C and 180 rpm for 20 – 22 h. The cultures were harvested by centrifugation (4000×*g*, 15 min, 4 °C), resuspended in HEPES buffer (20 mM HEPES, 200 mM NaCl, 20 mM KCl, 40 mM imidazole, pH 8.0), and subsequently disrupted using a microfluidizer (M110-L, Microfluidics). The cell debris was removed by centrifugation (50,000×*g*, 20 min, 4 °C). The supernatant was loaded onto Ni-NTA FF-HisTrap columns (GE Healthcare) for affinity purification via the hexahistidine tag. The columns were washed with HEPES buffer (10x column volume) and eluted with HEPES buffer containing 250 mM imidazole. The protein was concentrated with Amicon Ultra-30K centrifugal filters and subjected to SEC using a HiLoad 26/600 Superdex 200 column equilibrated in HEPES buffer without

imidazole and a pH of 7.5. The peak fractions were analyzed using a standard SDS-PAGE protocol, pooled, and concentrated with Amicon Ultra-30K centrifugal filters.

**Crystallization and structure determination.** Crystallization of MmCheY:-CheF_{CTD} was performed by the sitting-drop method at 20 °C in 0.5 µl drops consisting of equal parts of protein and precipitation solutions. Prior to crystallization, 2 mM BeF₂ and 20 mM NaF as well as 2U of thrombin were added to MmCheY:CheF_{CTD}. MmCheY:CheF_{CTD} crystallized at 800 µM concentration

within 24 h days in 0.2 MgCl$_2$, 0.1 M Tris pH 8.5 and 20 % (w/v) PEG 8000. Prior data collection, crystals were flash-frozen in liquid nitrogen employing a cryo-solution that consisted of mother-liquor supplemented with 30 % glycerol. Data were collected under cryogenic conditions at the European Synchrotron Radiation Facility at beamline ID30B using MxCube3[61]. Crystallization of MjCheF was performed by the sitting-drop method at 20 °C in 0.5 µl drops consisting of equal parts of protein and precipitation solutions at a concentration of 30 mg/ml of MjCheF. Crystals appeared in 0.08 M Na-Acetate pH 4.6, 1.6 M ammonium sulfate, 20% Glycerol after 2 weeks. Prior data collection, crystals were flash-frozen in liquid nitrogen employing a cryo-solution that consisted of mother-liquor supplemented with 30% glycerol. Data were collected under cryogenic conditions at BESSY beamline MX14.2 using MxCube2.

The data of PhCheF were retrieved from the Zenodo science data archive (https://doi.org/10.5281/zenodo.1148967)[46]. Data were integrated and scaled with XDS[62] and merged with XSCALE[62]. The structures of PhCheF and MjCheF were determined by selenium single-anomalous dispersion (Se-SAD) using the Phenix-implemented AutoSol program[63]. The structure of MmCheY:CheF was solved by molecular replacement with PHASER[64] using the structure of activated CheY (PDB: 6EKH) as search model. All structures were manually built in COOT[65], and iteratively refined with PHENIX[63]. Figures were prepared with PYMOL[66] and Chimera[67].

**Glutathione-S-transferase binding assays**. GST interaction assays were performed with SEC buffer (20 mM HEPES-Na (pH 7.5), 200 mM NaCl, 20 mM KCl, 20 mM MgCl$_2$) + 0.05% Tween at 4 °C using mobicol "classic" spin columns (MoBiTec). A total amount of 2 nmol of SEC-purified GST-tagged protein was immobilized on 25 µl Glutathione Sepharose (GE Healthcare) and incubated on a turning wheel for 5 min. Two equivalents of putative interaction partner proteins were added to the beads and incubated for 20 min on a turning wheel. After removal of residual protein by centrifugation (4 °C, 5000×g, 1 min), the column was washed three times with SEC buffer + 0.05% Tween. Proteins were eluted with 80 µl of GSH elution buffer (20 mM HEPES PH 8.0, 200 mM NaCl, 20 mM KCl, 0.05% Tween, 20 mM glutathione) and analyzed by Coomassie-stained SDS-PAGE. The experiment was performed three times independently.

**Isothermal titration calorimetry**. Prior to the measurement, the MmCheY2 and GST-MmCheF$_{\Delta NTD}$ protein solutions were dialyzed against the identical buffer, which consisted of 20 mM HEPES-Na (pH 7.5), 200 mM NaCl, 20 mM KCl, 20 mM MgCl$_2$, 20 mM NaF, and 2 mM BeF$_2$. Titration was carried out at a temperature of 25 °C with a MicroCal ITC200 (Malvern Panalytical Ltd). 280 µL of GST-MmCheF$_{\Delta NTD}$ at 25 µM were placed in the sample cell and the syringe was fully loaded with 25 µM of MmCheY2. The first injection of 0.3 µL was followed by 13 injections of 2 µL to generate the thermogram representing the interaction. The experiments are repeated in the absence of NaF and BeF$_2$. Data were processed with the MicroCal PEAQ-ITC Analysis Software (Malvern Panalytical Ltd).

**Mass photometry**. MP experiments were performed using a OneMP mass photometer (Refeyn Ltd, Oxford, UK). Data acquisition was performed using AcquireMP (Refeyn Ltd. v2.3). MP movies were recorded at 1 kHz, with exposure times varying between 0.6 and 0.9 ms, adjusted to maximize camera counts while avoiding saturation. Microscope slides (70 × 26 mm) were cleaned 5 min in 50% (v/v) isopropanol (HPLC grade in Milli-Q H$_2$O) and pure Milli-Q H$_2$O, followed by drying with a pressurized air stream. Silicon gaskets to hold the sample drops were cleaned in the same manner fixed to clean glass slides immediately prior to measurement. The instrument was calibrated using NativeMark Protein Standard (Thermo Fisher) immediately prior to measurements. Immediately prior to MP measurements, protein stocks were diluted directly in HEPES buffer. Typical working concentrations of MmCheF were 25–50 nM for the actual measurement. Each protein was measured in a new gasket well (i.e., each well was used once). To find focus, 18 µl of fresh room temperature buffer was pipetted into a well, the focal position was identified and locked using the autofocus function of the instrument. For each acquisition, 2 µL of diluted protein was added to the well and thoroughly mixed. The data were analyzed using the DiscoverMP software.

**Growth and genetic manipulation of H. volcanii**. H. volcanii strains were grown and genetically manipulated as described previously[26,45]. Transformation was performed with PEG 600 as described[68], and selected based on uracil in ΔpyrE2 strains. Strains are listed in table S4 The cells were cultured at 45 °C, unless specified otherwise, under constant rotation at 120 rpm, in complete YPC medium containing 5% Bacto$^{TM}$ yeast extract (BD Biosciences, UK), 1% peptone (Oxoid, UK), 1% Bacto$^{TM}$ Casamino acids or in selective CA medium containing 5% Bacto$^{TM}$ Casamino acids in 18% SW (Salt water, containing per liter 144 g NaCl, 21 g MgSO$_4$ X 7H$_2$O, 18 g MgCl$_2$ X 6H$_2$O, 4.2 g KCl, and 12 mM Tris HCl, pH 7.3).

**Western blotting**. To confirm the integrity of our GFP phusion constructs Western-Blots were performed. H. volcanii ΔcheF cells transformed with the corresponding expression plasmids were grown in 20 ml CA medium to an OD600 of 0.3 to get reliable Western-Blot signals. One hour before cells were harvested, protein expression was induced by the addition of 1 mM tryptophan. Cells were harvested by centrifugation at 3000×g for 20 min. The pellets were resuspended in 1x phosphate-buffered saline (137 mM NaCl, 2.7 mM KCl, 10 mM Na$_2$HPO4, 1.8 mM KH$_2$PO4 adjusted to PH 7.2) supplemented with 2.5 mM MgCl$_2$ and 10 µg/ml DNase I to a theoretical OD600 of 10. To lyse cells 0.1 % DDM (n-dodecyl β-D-maltoside) was added and the cells incubated on ice for 10 min. To analyze the cell lysates samples were mixed with SDS-loading buffer, boiled for 10 min and 10 µl per sample loaded on a 15 % SDS gel. Gels were either stained with Ready BlueTM (Sigma-Aldrich) protein gel stain or blotted on poly-vinylidenfluorid (PVDF) membranes. The GFP antibody (1:5000, produced in rabbit, Sigma-Aldrich SAB4301138) was incubated on the membranes over-night and incubated shaking at 4 °C. The next day the secondary antibody (1:10,000, anti-rabbit, horseradish peroxidase (HRP) coupled, Sigma-Aldrich A0545) was added and incubated on the membrane for 3 h. Western-Blot signals were taken with the iBright FL1500 system (Invitrogen).

**Motility assays of H. volcanii on semisolid agar plates**. Motility assays were performed as previously described[26,45]. Semi solid agar plates were made from YPC medium containing 0.3% agar, 50 µg/mL uracil, and 1 mM tryptophan. Cells were inoculated in 5 mL CA medium with 50 µg/mL uracil when required and grown over night until an OD of ~0.5. Drops of 10 µL of culture of each strain were used to inoculate semi solid agar plates. The experiment was performed at least 3 independent times and included 3 technical replicates per experiment. The motility rings were scanned after 5 days of incubation at 45 °C, and the diameters were measured.

**Fluorescence microscopy**. Fluorescence microscopy was performed as previously described[45]. In short, H. volcanii cells were grown in 5 mL CA medium and the next day the cultures were diluted to a theoretical OD of 0.005 in 20 mL of CA medium. After 16 h incubation at 42 °C, the cultures typically reached an OD of 0.01–0.05 and were imaged. During the last hour before observation by microscopy, 0.2 mM tryptophan was added to the medium. For imaging, cells were spotted on an agarose pad made of 1% agar in 18% SW. The cells were observed at ×100 magnification in the phase contrast (PH3) mode on a Zeiss Axio Observer 2.1 Microscope equipped with a heated XL-5 2000 Incubator running VisiVIEW® software. Each experiment was repeated at least three independent times resulting in the analysis of over 500 cells per strain.

**Image analysis**. Microscopy images were analyzed with the ImageJ plugin MicrobeJ[69]. The number of fluorescent foci per cell was determined, and the cells were binned based on the number of intracellular foci. The number of cells with the same number of foci was calculated as a percentage of the total number of cells. To determine if the percentage of cells with foci was significantly different between strains, an unpaired two-tailed T-test was performed on the percentages calculated for each independent experiment (minimally 3). Total number of analyzed cells was >500 per strain.

**Time lapse microscopy and cell tracking**. Cells were grown as described for the fluorescence microscopy. In all, 1 mL of each culture was placed in a round DF 0.17 216 mm microscopy dish (Bioptechs) and observed at 40x magnification in the PH2 mode with a Zeiss Axio Observer 2.1 Microscope equipped with a heated XL-5 2000 Incubator heated to 45 °C running VisiVIEW® software. The swimming trajectories of the cells in recorded 15 s time lapse movies were determined using Visiview and Metamorph as previously described[26].

**Statistics and reproducibility**. All experiments were at least repeated three times independent of each other. No statistical method was used to predetermine sample size. No data were excluded from the analyses. The experiments were not randomized. The Investigators were not blinded to allocation during experiments and outcome assessment.

## Data availability

The coordinates and structure factors generated in this study have been deposited in the PDB database under accession codes PDB-7OD9 and PDB-7OVP. All other data generated in this study are provided in the Supplementary Information or Source Data file. Source data are provided with this manuscript.

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

## Acknowledgements

We thank Paithankar and coworkers for sharing their crystallographic data via the Zenodo database. We thank Florian Kraus for kindly providing us with BeF$_2$. We acknowledge Georg Hochberg for access to the Refeyn OneMP. We are grateful for excellent beamline access and support by the European Synchrotron Radiation facility (ESRF). F.A. thanks the Peter und Traudl Engelhorn foundation for financial support. G.B. thanks the Deutsche Forschungsgemeinschaft (DFG) for support through the Collaborative Research Council TRR174. T.E.F.Q. is supported by the DFG via an Emmy Noether grant (411069969). P.N. was supported by a VW foundation grant (Momentum, grant number 94933) and S.V.A. received funding from the SFB1381 (German Research Foundation 403222702-SFB1381). Z.L. was supported by a CSC fellowship. G.B. thanks the Max Planck Society for support.

## Author contributions

F.A., T.E.F.Q., S.V.A., and G.B. conceived of the project and designed the study. F.A. and T.E.F.Q. wrote the paper. F.A, T.E.F.Q., P.W., P.N., P.I.G., M.P., and Z.L. performed experiments. F.A., T.E.F.Q., P.I.G., and M.G. analyzed data. T.E.F.Q., D.O., S.V.A., and G.B. contributed funding and resources. All authors read and commented on the manuscript.

## Funding

## Competing interests

The authors declare no competing interests.
