## [Peer Review File · Nature Communications]

REVIEWER COMMENTS

Reviewer #1 (Remarks to the Author):

This paper reports the structure of full-length CheF and a C-terminal fragment of CheF in complex with phosphorylated CheY (CheY-P). The archaeal flagellar system is very different from the bacterial flagellar system. However, both systems share a similar chemotaxis signaling mechanism in which the binding of CheY-P to the flagellar basal body induces switching of direction of flagellar rotation. CheF is a unique component of the archaeal flagellar system and is involved in chemotaxis switching of archaeal flagellum. This paper is the first report that describes the structural basis of the CheY-P recognition by CheF. The authors found that CheF form a domain swapped dimer and clearly showed that the C-terminal domain of CheF is responsible for the CheY binding and the N-terminal domain is involved in interaction with the archaeal flagellar C-ring. On the basis of the structures and mutation analyses, the authors propose a model how CheY-P binding transmit to the basal body complex via CheF.

This paper provides new insight on the switching mechanism of the archaeal flagellar motor. However, I found some issues that should be clarified before publication.

Specific comments

(1) The authors used proteins with the N-terminal GST-tag for the CheY binding assay (Fig. 2). GST forms a dimer and its two C-termini locate in opposite side of the dimer. Therefore, the CheF chains are present in the opposite side of the GST dimer. Can the CheF chains form a homo-dimer that was found in the crystal structure on the GST dimer?

(2) How does the 'twist' occur? What is the key residue and conformation to form the twisted dimer?

(3) No description about the MjCheF structure. MjCheF forms a domain swapped dimer? If so, what is the relative orientation of the two subunits? What is the sequence similarity? The MjCheF structure is useful to assess the significance of the twisted dimer as well as the domain swapped nature of CheF, even though the resolution is much lower than that of PhCheF.

(4) Line 353-, section "The CTD of CheF is important for directional movement of *H. volcanii*": Δ CheF:: wt-cheF should be examined as a control in Fig 3b, c, and d. Expression level of the CheF mutants should also be shown.

(5) Line 431: Mw derived from MALS is an absolute molecular mass. The authors should show radius of gyration derived from the SEC-MALS measurements.

(6) Line 482-483 and Line 490-491: The authors argue that the binding site of FliMm for CheY-P is similar to that of CheF, but I could not find the similarity. Maybe an additional figure will help the understanding.

(7) Line 52-9- Section "CheY-P binding leads to a conformational change at CheF" and Fig. 6b and C: The author stated that the conformational change of CheF induced by the CheY-P binding is mostly limited to alpha 8 in line 448. What conformational change led to the displacement of the PH domains? The structure of the CheF fragment with CheY-P does not have the PH-domain nor the N-terminal half of alpha4. How did the author find the movement of the PH-domain upon CheY binding? I think that it is impossible to estimate the orientation of the PH domain in the CheY-P bound state from their data.

Other minor comments

(8) Line 32, "This mechanism...": This sentence is duplicated to the former sentence.

(9) No description about Se-Met MjCheF in "Materials and Methods". Expression, purification, crystallization, data collection, and structure determination of Se-Met MjCheF should be described.

(10) Line 161: Purification of GST-MmCheF constructs should be described.

(11) Line 173: Did the authors crystallize MmCheY:CheFCTD with thrombin?

(12) Line 180-181, "The structure of PhCheF was determined by selenium single-anomalous dispersion": This sentence may be wrong. In the main text, the authors described that PhCheF is solved by MR using the model of MjCheF.

(13) Line 324, "FigS2a": S2b. Fig S2a is not referred in the text.

(14) Line 336, "compare also to": Compare to what?

(15) Line 401, "in the absence if": "in the absence of"

(16) Line 403, "ΔpyrE2::pTA1228": Use the same terminology throughout the document.

Reviewer #2 (Remarks to the Author):

The manuscript determined the atomic structures of CheF alone and in complex with activated CheY by X-ray crystallography. CheF forms an elongated dimer with a twisted architecture. The manuscript showed that CheY binds to the C-terminal tail domain of CheF leading to slight conformational changes within CheF. The structural, biochemical and genetic analyses reveal the mechanistic basis for CheY-P binding to CheF and allow the authors to propose a model for rotational switching of the archaellum. The manuscript provided useful information about the interaction mechanism of CheF and CheY-P, and the possible mechanism of CheY-P induced rotational switching in the archaellum. However, some important analysis in the manuscript are flawed and should be improved upon.

Major:

The interpretation of the results of the t-test are incorrect and misleading. For example, Fig.3c, "****p>0.01**" means there is not much difference between full-length GCP-CheF and CTD-deleted CheF, in contradiction to that claimed in the main text.

Fig. 3d, "****p>0.01**" means not much difference. e.g., 0.6 vs 0.8 reversals per second is not "a significant reduction" noting the large error bars in the data of Fig. 3d.

Therefore, the conclusion that "The CTD of CheF is important for directional movement" is invalid.

Minor:

1. line 398: "interaction with CheF" -> "interaction with CheY-P".

2. line 401: if -> of

3. Explain CheF_CTD at its first occurrence.

4. line 485 and in the introduction: at least in E. coli and Salmonella, FliM_M does not have phosphatase activity and does not rapidly dephosphorylate CheY-P. Please do not generate the results of some specific bacterial species to all bacteria.

Reviewer #3 (Remarks to the Author):

Background

This manuscript covers the late stages of chemotaxis in a thermophilic organism. The authors produced and determined the structure of a fusion between CheF and CheY and activate CheY via beryllium fluoride, a mimic of the phosphorylated state. ITC was used to quantify affinity. The mode of binding between CheY and CheF was compared and contrasted with CheY and FliM from bacteria.

Significance

This represents an advance in our understanding of chemotaxis in the archaea sufficient to warrant publication. The final steps in chemotaxis have been somewhat recalcitrant to complete elucidation, and more information is always welcome. The difference between bacteria and archaea is surprising.

Abstract

The abstract is sufficient, but it would benefit from slightly less background information.

Introduction

This section of the manuscript is clearly written.

Lines 125-126.

What does well-designed mean? Perhaps a different choice of words would be better.

Methods

The experiments are described in sufficient detail to allow them to be replicated.

Results and Discussion

Beginning at line 386

Why does the GFP-CheFCTD(Δ) behave differently from the Δ (α 8) fusion protein? In other words why is the latter unstable when the former is not?

Lines 500-501

"Furthermore, the affinity of archaeal CheY-P towards CheF is significantly higher (1.24 μ M) than that of bacterial CheY-P to FliM (\sim 27 μ M of FliMN towards CheY 39,66), while the affinity towards FliMM is probably even lower."

There are other comparisons that seem useful. The affinity between CheY and FliMNM is 1.7 μ M, and the affinity between CheY-BeF3 and FliMMN is 39 nM (Park 2006).

Typographical errors

Instable should be unstable. (line 389)

Figures and Figure Captions

I would include the presence of beryllium fluoride in the caption of Figure 2b, but this is a small matter.

Supplementary information

All is in order.

Reviewer #1 (Remarks to the Author):

This paper reports the structure of full-length CheF and a C-terminal fragment of CheF in complex with phosphorylated CheY (CheY-P). The archaeal flagellar system is very different from the bacterial flagellar system. However, both systems share a similar chemotaxis signaling mechanism in which the binding of CheY-P to the flagellar basal body induces switching of direction of flagellar rotation. CheF is a unique component of the archaeal flagellar system and is involved in chemotaxis switching of archaeal flagellum. This paper is the first report that describes the structural basis of the CheY-P recognition by CheF. The authors found that CheF form a domain swapped dimer and clearly showed that the C-terminal domain of CheF is responsible for the CheY binding and the N-terminal domain is involved in interaction with the archaeal flagellar C-ring. On the basis of the structures and mutation analyses, the authors propose a model how CheY-P binding transmit to the basal body complex via CheF.

This paper provides new insight on the switching mechanism of the archaeal flagellar motor. However, I found some issues that should be clarified before publication.

Specific comments

(1) The authors used proteins with the N-terminal GST-tag for the CheY binding assay (Fig. 2). GST forms a dimer and its two C-termini locate in opposite side of the dimer. Therefore, the CheF chains are present in the opposite side of the GST dimer. Can the CheF chains form a homo-dimer that was found in the crystal structure on the GST dimer?

We thank this reviewer for this important notion. The linker between the N-terminal GST and the CheF-CTD allows a sufficient flexibility for GST- and CheF dimerization. We have now included mass photometry data on GST-CheFdNTD showing that there is higher oligomeric species indicating additional oligomerization through GST (Fig. S3b). However, the main fraction is dimeric. We assume that the CheF dimerization forces GST into a monomeric configuration, while it usually is present in a monomer-dimer equilibrium. We included the following paragraph in the text: "Here, we employed the GST-MmCheF_{CTD} construct used for the interaction assays at it showed a higher stability compared to a construct lacking the GST-tag. As GST forms dimers, we investigated a potential oligomerization of the fusion protein by mass photometry. Our data show that the main fraction (74%) has a molecular weight of 76 kDa indicating a dimer, while only 20% of the molecules had a molecular weight of 152 kDa indicating a tetramer."

(2) How does the 'twist' occur? What is the key residue and conformation to form the twisted dimer?

That is indeed an interesting question. We now included panel d in figure 1 showing a superposition of the two monomers of the dimer showing the conformational flexibility of the two domains in respect to each other. The describing paragraph reads as follows: "Superposition of the two monomers using N-terminal residues 2-260 (r.m.s.d = 0.8 Å) shows this rotational movement of the CTD (Fig. 1d). The residues involved in bending are E265 to L267 within helix $\alpha 8$ (obtained from DynDom⁵⁸). This observation suggests that the CheF homodimer exhibits an intrinsic flexibility with respect to its N-terminal domains, which might be of functional relevance."

(3) No description about the MjCheF structure. MjCheF forms a domain swapped dimer? If so, what is the relative orientation of the two subunits? What is the sequence similarity? The MjCheF structure is useful to assess the significance of the twisted dimer as well as the domain swapped nature of CheF, even though the resolution is much lower than that of PhCheF.

MjCheF represents a special case among CheF homologs, as it is lacking part of the CTD that we identified crucial for CheY-P binding and dimerization. This 'short' CheF is only found in 3 of 147 known CheF sequences. Our dataset(s) of MjCheF unfortunately suffer from a strong tNCS that prevents finalizing the MjCheF model for deposition. However, we would like to share our data confidentially with this reviewer (see figure below) as we are still investigating this special type of CheF for a future study. The overall architecture of the N-terminal PH-domains of MjCheF is very similar to the ones of PhCheF (r.m.s.d. 1.2Å), while the CTD is absent from helix $\alpha 4$ on. Interestingly, MjCheF still seems to form dimers as judged by mass photometry (main fraction of 59 kDa). We currently have no experimental evidence if this type of CheF is functional and actually capable of CheY-binding. However, if that is the case, nature has evolved a fundamentally different mechanism of CheY-P signal transduction towards CheF in Methanococcus jannaschii. We hope that this reviewer agrees that this clearly is beyond the scope of this manuscript and should rather be investigated in detail in a separate study.

(4) Line 353-, section “The CTD of CheF is important for directional movement of *H. volcanii*”: Δ cheF:: wt-cheF should be examined as a control in Fig 3b, c, and d. Expression level of the CheF mutants should also be shown.

To give insight into the expression level of the several CheF mutants in Haloferax volcanii, we have performed western blot, to show that expression levels of all constructs are equal. This blot is added to the article as figure S4. In addition, we have added our western blotting procedure in the material and methods section.

(5) Line 431: Mw derived from MALS is an absolute molecular mass. The authors should show radius of gyration derived from the SEC-MALS measurements.

Unfortunately, our SEC-MALS setup only allows a reliable calculation of the particle diameter of molecules larger than 10 nm, which is not the case for CheY:CheF. The differences in calculated masses arise from a higher standard deviation in the sample without BeF. We have rephrased the paragraph accordingly: “Interestingly, the mass determination was more accurate and resulted in a determined molecular weight of 50 ± 5 kDa (Fig. S5a, right panel).”

(6) Line 482-483 and Line 490-491: The authors argue that the binding site of FliMm for CheY-P is similar to that of CheF, but I could not find the similarity. Maybe an additional figure will help the understanding.

Thank you for pointing this out. The binding site of FliM-M to CheY-P was poorly described in the original publication. We have highlighted the regions they identified to be involved in FliM_M interaction in Fig. 5.

(7) Line 52-9- Section “CheY-P binding leads to a conformational change at CheF” and Fig. 6b and C: The author stated that the conformational change of CheF induced by the CheY-P binding is mostly limited to alpha 8 in line 448. What conformational change led to the displacement of the PH domains? The structure of the CheF fragment with CheY-P does not have the PH-domain nor the N-terminal half of alpha4. How did the author find the movement of the PH-domain upon CheY binding? I think that it is impossible to estimate the orientation of the PH domain in the CheY-P bound state from their data.

We completely agree that our presentation of the conformational change at CheF was poorly described and not justified by our data. We have now included a more thorough analysis of the conformational changes at CheF induced by CheY-P binding and combined it with an analysis of the inherent flexibility of the N-terminal PH-domains towards the CTD as deduced from the CheF dimer (Fig. 1d). The paragraph now reads: “As our CheF structures in activated and non-activated states were derived from two different organisms, we first investigated the similarity of the CTD’s (Fig. 6b). Despite the sequence similarity of 36 %, the domains showed almost identical structural elements (Fig. 6b). Upon superposition, we observed some slight conformational changes as helix a8 was pulled towards the phosphorylation site (Fig. 6c). In both structures, helix a8 and a4 of the opposing monomers are tightly connected via a hydrophobic interface and polar contacts (Fig. 6c). Thus, the movement of helix a8 displaces helix a4. As our structure of CheF bound to an activated CheY is lacking the N-terminal PH domains, we can only hypothesize that this movement is further transmitted. However, combined with our observation of the inherent flexibility of the CTD’s through a

patch in helix $\alpha 4$ (compare to Fig. 1d), we consider it likely that slight conformational changes upon CheY-binding in the CTD would lead to larger changes in the PH-domains.” Although we can not predict the exact displacement of the PH domains, we hope that this reviewer shares our conclusion that the displacement of helix $\alpha 4$ has to have implications on the relative positioning of the N-terminal PH-domains towards the CTD.

Other minor comments

(8) Line 32, “This mechanism...”: This sentence is duplicated to the former sentence.

We have changed this sentence.

(9) No description about Se-Met MjCheF in “Materials and Methods”. Expression, purification, crystallization, data collection, and structure determination of Se-Met MjCheF should be described.

Has been added.

(10) Line 161: Purification of GST-MmCheF constructs should be described.

The GST-MmCheF constructs were purified via the same purification scheme as the other proteins. Our GST carries an N-terminal hexahistidine-tag that was used for IMAC.

(11) Line 173: Did the authors crystallize MmCheY:CheFCTD with thrombin?

Yes indeed. We added thrombin to the protein solution prior to crystallization as outlined in the materials and methods section.

(12) Line 180-181, “The structure of PhCheF was determined by selenium single-anomalous dispersion”: This sentence may be wrong. In the main text, the authors described that PhCheF is solved by MR using the model of MjCheF.

This sentence is correct, but it obviously is misleading. We initially were able to solve the PhCheF structure with our initial MjCheF model but we could also get a substructure from the selenium sites. We finally combined SAD phases and initial MR phases for complete model building.

(13) Line 324, “FigS2a”: S2b. Fig S2a is not referred in the text.

Has been added.

(14) Line 336, “compare also to”: Compare to what?

The citation has been added.

(15) Line 401, “in the absence if”: “in the absence of”

Has been changed.

(16) Line 403, "ΔpyrE2::pTA1228": Use the same terminology throughout the document.

Has been changed.

Reviewer #2 (Remarks to the Author):

The manuscript determined the atomic structures of CheF alone and in complex with activated CheY by X-ray crystallography. CheF forms an elongated dimer with a twisted architecture. The manuscript showed that CheY binds to the C-terminal tail domain of CheF leading to slight conformational changes within CheF. The structural, biochemical and genetic analyses reveal the mechanistic basis for CheY-P binding to CheF and allow the authors to propose a model for rotational switching of the archaellum. The manuscript provided useful information about the interaction mechanism of CheF and CheY-P, and the possible mechanism of CheY-P induced rotational switching in the archaellum. However, some important analysis in the manuscript are flawed and should be improved upon.

Major:

The interpretation of the results of the t-test are incorrect and misleading. For example, Fig.3c, "***p>0.01" means there is not much difference between full-length GCP-CheF and CTD-deleted CheF, in contradiction to that claimed in the main text.

In the figure legend of 3c, was a typo in the P-value that we have now corrected to: 'P<0.0001'. This is a significant difference.

Fig. 3d, "***p>0.01" means not much difference. e.g., 0.6 vs 0.8 reversals per second is not "a significant reduction" noting the large error bars in the data of Fig. 3d. Therefore, the conclusion that "The CTD of CheF is important for directional movement" is invalid.

The exact values of the T-test are:

H26 + pTA1228 vs. dcheF + pGFP-CheF: p = 0.371 -> ns

*dcheF + pGFP-CheF vs. dcheF + pTA1228: p < 0.0001 -> *****

*dcheF + pGFP-CheF vs. dcheF+ GFP-CheFdC: p < 0.0001 -> *****

To clarify that these differences are indeed significant, we have mentioned now these exact values in the figure legend. In addition, we have added the two instead of one digit values to the text (instead of writing ~0.6 vs 0.8 reversals per second to 0.82 vs 0.55 reversals per second). For clarity: figure 3d does not contain 'error bars', instead we show here a box and whisker plot to present in one view the raw data. We presume that the whiskers were taken as very large error bars by the reviewer. We regret this confusion.

Minor:

1. line 398: "interaction with CheF" ->"interaction with CheY-P".

Has been changed.

2. line 401: if -> of

Has been changed.

3. Explain CheF_CTD at its first occurrence.

Has been done.

4. line 485 and in the introduction: at least in E. coli and Salmonella, FliM_M does not have phosphatase activity and does not rapidly dephosphorylate CheY-P. Please do not generate the results of some specific bacterial species to all bacteria.

We have changed the sentences accordingly.

Reviewer #3 (Remarks to the Author):

Background

This manuscript covers the late stages of chemotaxis in a thermophilic organism. The authors produced and determined the structure of a fusion between CheF and CheY and activate CheY via beryllium fluoride, a mimic of the phosphorylated state. ITC was used to quantify affinity. The mode of binding between CheY and CheF was compared and contrasted with CheY and FliM from bacteria.

Significance

This represents an advance in our understanding of chemotaxis in the archaea sufficient to warrant publication. The final steps in chemotaxis have been somewhat recalcitrant to complete elucidation, and more information is always welcome. The difference between bacteria and archaea is surprising.

Abstract

The abstract is sufficient, but it would benefit from slightly less background information.

Introduction

This section of the manuscript is clearly written.

Lines 125-126.

What does well-designed mean? Perhaps a different choice of words would be better.

We have removed this description.

Methods

The experiments are described in sufficient detail to allow them to be replicated.

Results and Discussion

Beginning at line 386

Why does the GFP-CheFCTD(delta) behave differently from the delta(alpha8) fusion protein?
In other words why is the latter unstable when the former is not?

That is an interesting question. We assume that the boundaries of helix a8 are slightly different between Ph/MmCheF and the Haloferax homolog. This difference probably leads to an unstable protein construct in HvCheF.

Lines 500-501

“Furthermore, the affinity of archaeal CheY-P towards CheF is significantly higher (1.24 μM) than that of bacterial CheY-P to FliM ($\sim 27 \mu\text{M}$ of FliMN towards CheY 39,66), while the affinity towards FliMM is probably even lower.”

There are other comparisons that seem useful. The affinity between CheY and FliMNM is 1.7 μM , and the affinity between CheY-BeF3 and FliMMN is 39 nM (Park 2006).

*Thank you for pointing this out. We have included these data & rewritten the paragraph. It now reads: “In the flagellar system different affinities of CheY-P towards FliM were reported. While Park and coworkers reported a K_d of $39 \pm 5 \text{ nM}$ of activated CheY towards FliM_{NM} (residues 1-249) in *T. maritima*⁶⁷, McAdams et al. only observed a K_d of 27 μM of activated CheY towards a FliM_N peptide³⁹. The affinity of archaeal CheY-P towards CheF of 1.24 μM is between those observed in the flagellar systems.”*

Typographical errors

Instable should be unstable. (line 389)

Has been changed.

Figures and Figure Captions

I would include the presence of beryllium fluoride in the caption of Figure 2b, but this is a small matter.

Has been added.

Supplementary information

All is in order.

REVIEWERS' COMMENTS

Reviewer #1 (Remarks to the Author):

The revised version of the manuscript fully addressed my concerns.

Reviewer #2 (Remarks to the Author):

The authors addressed most of my comments, except that some corrects that were claimed to be made (in the rebuttal) actually did not appear in the manuscript. Please revise them. I have no further comment:

1. "In the figure legend of 3c, was a typo in the P-value that we have now corrected to: 'P<0.0001'. This is a significant difference."

The authors did not correct this in the figure legend of 3c. It still shows: "***P>0.01 as calculated with T-test".

2. "The exact values of the T-test are:

H26 + pTA1228 vs. dcheF + pGFP-CheF: $p = 0.371$ -> ns

dcheF + pGFP-CheF vs. dcheF + pTA1228: $p < 0.0001$ -> ****

dcheF + pGFP-CheF vs. dcheF+ GFP-CheFdC: $p < 0.0001$ -> ****

To clarify that these differences are indeed significant, we have mentioned now these exact values in the figure legend."

Traditionally people would use '****' to represent $P < 0.0001$ (and '**' for $P < 0.01$) in the figure. I would recommend using **** instead of ** in fig. 3c & d to avoid confusion, and revise the figure legend accordingly.

Reviewer #3 (Remarks to the Author):

The authors have addressed all of the points I raised in the original submission.

I found two small matters of concern in the resubmission. In Figure 2b it is unclear what the solid line represents. It appears to connect the data points with straight line segments, as opposed to being a smooth curve. Ideally the authors would use the same fitted curve used to find the value of K_d obtained from nonlinear regression. In any event, exactly what the line represents should be clarified.

In Figure 6b I do not understand the color coding system for the one letter representations of the amino acids. This sentence is part of the caption of Figure S1b: "Color coding according to the physico-chemical properties of the amino acids." Could this sentence be added into the caption of 6b? Also I noticed that in both figures the same amino acid was sometimes given two colors. In Figure S1b some D's are black, and others are purple. The Ks are either black or red. The H's are either black or blue. In Figure 6b Some K's are black and some are red. It would be helpful for there to be consistency in the color coding.

Reviewer #2 (Remarks to the Author):

The authors addressed most of my comments, except that some corrects that were claimed to be made (in the rebuttal) actually did not appear in the manuscript. Please revise them. I have no further comment:

1. "In the figure legend of 3c, was a typo in the P-value that we have now corrected to: 'P<0.0001'. This is a significant difference."

The authors did not correct this in the figure legend of 3c. It still shows: "***P>0.01 as calculated with T-test".

2. "The exact values of the T-test are:

H26 + pTA1228 vs. dcheF + pGFP-CheF: $p = 0.371$ -> ns

dcheF + pGFP-CheF vs. dcheF + pTA1228: $p < 0.0001$ -> ****

dcheF + pGFP-CheF vs. dcheF+ GFP-CheFdC: $p < 0.0001$ -> ****

To clarify that these differences are indeed significant, we have mentioned now these exact values in the figure legend."

Traditionally people would use '****' to represent $P < 0.0001$ (and '***' for $P < 0.01$) in the figure. I would recommend using **** instead of ** in fig. 3c & d to avoid confusion, and revise the figure legend accordingly.

We excuse for not having updated the figure legend accordingly. We have now changed figure and legend as suggested by this reviewer.

Reviewer #3 (Remarks to the Author):

The authors have addressed all of the points I raised in the original submission.

I found two small matters of concern in the resubmission. In Figure 2b it is unclear what the solid line represents. It appears to connect the data points with straight line segments, as opposed to being a smooth curve. Ideally the authors would use the same fitted curve used to find the value of K_d obtained from nonlinear regression. In any event, exactly what the line represents should be clarified.

In Figure 6b I do not understand the color coding system for the one letter representations of the amino acids. This sentence is part of the caption of Figure S1b: "Color coding according to the physico-chemical properties of the amino acids." Could this sentence be added into the caption of 6b? Also I noticed that in both figures the same amino acid was sometimes given two colors. In Figure S1b some D's are black, and others are purple. The Ks are either black or red. The H's are either black or blue. In Figure 6b Some K's are black and some are red. It would be helpful for there to be consistency in the color coding.

Thank you for pointing this out. We have now changed the solid line to a fitted curve and added a sentence to explain it in the figure caption: "The black dots represent the ΔH per injection of titrant into the cell and the solid line represents the fitting curve for all recorded injections."

We have changed the description in the figure legend: "Residues are colored according to the Clustal X coloring scheme that depends on the residue type and conservation pattern in the respective column. "